# Prevalence, locations and predictors of attitudes accepting both intimate partner violence and additional forms of violence against women and girls in South Sudan: a geospatial analysis

Angelo Lamadrid[1], Ignacio Leiva-Escobar[2], Caroline Jeffery[1,3], Robert J. Anguyo[1], Richard Lako[4], Joseph J. Valadez[1]*

1 Department of International Public Health, Liverpool School of Tropical Medicine, Liverpool, United Kingdom, 2 Internal Medicine IX-Department of Clinical Pharmacology and Pharmacoepidemiology, Medical Faculty of Heidelberg, Heidelberg University, Heidelberg, Germany, 3 Institute of Infection, Veterinary and Ecological Science, University of Liverpool, Liverpool, United Kingdom, 4 Research, Monitoring and Evaluation, Ministry of Health, Juba, Republic of South Sudan

☯ Co-first authors.
* joseph.valadez@lstmed.ac.uk

## Abstract

Most research on violence against women and girls (VAWG) in South Sudan has focused on intimate partner violence (IPV) neglecting other forms of VAWG. This research aims to determine the prevalence of attitudes accepting IPV and whether it overlaps with attitudes accepting additional forms of VAWG (child marriage, raiding villages for women during cattle rustling or other raids or female genital mutilation) in South Sudanese men and women, or are different attitudinal phenomena. We used data from the National Household Survey South Sudan 2020 (n = 1,732 women, n = 1,730 men aged 15-49 years). We estimated attitudinal prevalences and applied spatial analysis (Global Moran's I, Getis and Ord's local Gi*, and Kuldorff's SatScan) and multilevel regression to assess overlapping attitudes accepting IPV and at least one other form of VAWG studied in the 10 states and three administrative areas comprising the country. The prevalence of attitudes accepting IPV overlapping with the prevalence of attitudes accepting at least one other form of VAWG was 34.72% (95% CI = 33.14%-36.34%). Sub-national results were non-randomly correlated (Global Moran's I= 0.23). Higher clusters displaying overlaps were located in the counties Kapoeta East, Kapoeta South, Kapoeta North, Budi, Pibor, and Ikotos. People married, cohabiting or living together (aOR = 1.40, 95% CI = 1.04-1.90) as well as people widowed, divorced or separated (aOR = 1.75, 95% CI = 1.05-2.93) were associated with attitudinal overlaps. Conversely, communities with any formal education were associated with a lower odds of overlapping (aOR = 0.26. 95% CI = 0.09-0.70). In South Sudan overlapping acceptance of IPV and at least one other form of VAWG are spatially clustered. Therefore, strategies to understand and tackle them should be strengthened especially in those locations. Essential elements include increasing schooling for children and promoting women's empowerment, especially among male-female partnerships. These conclusions have national and international policy implications.

**Data availability statement:** Data is available from UNICEF South Sudan which can be contacted through https://www.unicef.org/southsudan/contact.

**Funding:** The geospatial research was funded by UNITAID (STAR Initiative), sub-agreement number 4214-CeSHHAR. Professor Valadez was funded by the Liverpool School of Tropical Medicine. Funding for the South Sudan National Household Survey and its analysis was provided through a grant from UNICEF South Sudan (SSD/PCA2019614). The funders did not have any input into the conduct and analysis of the study, the writing of the manuscript nor the decision to submit the manuscript for publication.

**Competing interests:** The authors have declared that no competing interests exist.

## Introduction

Violence Against Women and Girls (VAWG) is a significant global public health issue and a violation of human rights. Although gender equality was included in the 1948 Universal Declaration of Human Rights and Sustainable Development Goal 5 aims to eliminate all forms of VAWG by 2030 [1], achieving this target remains elusive. For instance, the prevalence of Intimate Partner Violence (IPV), the most common form of VAWG, has remained static over the past decade [2,3]. Globally, 30% of women experience physical, sexual, and/or psychological violence by their partner [1].

The Causes of VAWG are complex [4]. An asymmetrical power distribution between men and women, characterized by the overvaluation of masculinity based on patriarchal norms, may be a significant factor [5–7]. Risk factors associated with VAWG include low education, poverty [1], religion, youth, disability, and residing in areas with attitudes that accept VAWG or affected by humanitarian crises or natural disasters [8,9].

Attitudes accepting VAW are social norms at institutional, community, and personal levels that justify the perpetration of violence against women and blame the victims themselves [10]. Although acceptance of VAWG does not necessarily mean that women and girls will experience it [5], there is a strong relationship between such attitudes and a higher prevalence of VAWG [4]. Particularly when analysed within an ecological [4,11] or socioecological framework [9].

For instance, in communities where women justify VAWG, they are 41% more likely to experience IPV compared to women in communities where such justification is not present [6]. Similarly, women in conflict-affected settings face a threefold higher risk of experiencing VAWG compared to women in non-conflicted areas [12,13].

South Sudan is among the countries with the highest prevalence of VAWG worldwide [14]. While the global prevalence of IPV is 27%, it is 41% in South Sudan [14]. In some states, the rates of physical and/or sexual violence rise to 65% [15]. Additionally, South Sudan has the seventh-highest prevalence rate of child marriage globally [16]. In sub-Saharan Africa and South Asia, approximately 40% of girls are married before the age of 18, whereas in South Sudan, this rate is 52% [16,17]. The World Health Organization (WHO) classifies South Sudan as a grade 3 emergency due to its unstable economic and political situation [18], and it ranks third in humanitarian needs according to the Fragile States Index [19], further exacerbating the situation for VAWG [20,21].

While the prevalence of VAWG in South Sudan has been studied [3,15,22], there is limited research on the relationship between attitudes toward IPV and other forms of VAWG common in the sub-Saharan region [23], such as child marriage [16,22], abduction of women during cattle raids [24], and female genital mutilation [22]. Furthermore, there is little information on the geographic distribution of these attitudes and whether they occur together or independently. Previous spatial analyses of VAWG have proven beneficial in identifying clusters and spatial determinants, which can help target resources and plan strategies to address VAWG in its various forms [25–33].

Our research aims to (i) determine the prevalence of attitudes accepting IPV, child marriage, abduction of women during cattle rustling or other raids, and female genital mutilation, as well as the prevalence of overlapping attitudes towards IPV and at least one other form of VAWG; (ii) analyse the distribution and clustering of these overlapping attitudes; and (iii) identify factors associated with these overlapping attitudes in South Sudan.

This study will assist national policymakers and international agencies in targeting interventions to priority areas in South Sudan, with the goal of reducing the acceptance of VAWG and, potentially, decreasing the prevalence of VAWG events.

## Materials and methods

### Setting

In 2011, after 20 years of civil war in Sudan, South Sudan became an independent nation and the newest African country [34]. It comprises 10 states: Central Equatoria (where the capital city, Juba, is located), Western Equatoria, Eastern Equatoria, Lakes, Warrap, Western Bahr el-Ghazal, Northern Bahr el-Ghazal, Upper Nile, Jonglei, and Unity, along with three administrative areas (AA): Greater Pibor, Abyei, and Ruweng [35]. Each state is divided into counties, totalling 79 [36].

Since its independence, South Sudan has been unstable due to oppositional military forces and ethnic disparities, even after signing a peace agreement in 2018 [34]. As a result, 2.2 million are internally displaced people, and over 2 million have sought refuge in bordering countries [21]. Additionally, climate change, extreme poverty, VAWG, and poor access to water, sanitation, and hygiene services have boosted the adverse conditions of life, leaving 9 million people in need of humanitarian assistance [21].

### Study design and sample size

This research is a cross-sectional study based on a secondary analysis of data from the National Household Survey (NHS) South Sudan 2020, specifically from the "Domestic and Gender-based Violence" questionnaire. The study design utilised the Lot Quality Assurance Sampling method (LQAS) which is an established method developed in the 1920s for quality control of industrial batch-production and adapted to healthcare settings during the mid-1980s that enables health system managers and researchers to classify a specific area based on a service delivery target [37].

By applying LQAS principles to this research, we treated the 79 counties as strata, and refer to them as supervision areas (SA) as they align with health system service delivery units [38–40]. However, the largest counties were divided into two parts, and the administrative areas were divided into four additional SA, resulting in a total of 89 SA [36].

In each SA of the 10 states, a random sample of 19 females and 19 males was estimated, and 24 females and 24 males in each of the three AA (S1 Table). These sample sizes were selected to ensure 95% confidence intervals did not exceed 0.10 when the SA level data were aggregated to the state level (e.g., n = 5 SA x 19 = 95, n = 4 SA x 24 = 96). Interview locations were sampled with probability proportional to size (PPS) using the population projection estimate for 2020 from the 2008 Census [35]. Interviews were selected with segmentation sampling [41]. Although the estimated sample size was 1,751 for each gender, it was not possible to complete data collection in Nyirol County (in Jonglei state) due to insecurity [35], resulting in a sample of 1,741 females and 1,739 males. Further information on the survey design and sampling method is described elsewhere [23,35].

Although data collection included the three administrative areas in South Sudan (Abyei, Greater Pibor, and Ruweng), in our analyses, Greater Pibor AA was merged with Jonglei and Ruweng AA with Unity to coincide with the Subnational Administrative Boundaries published by the United Nations [42]. Additionally, we excluded Nyirol county from our analysis due to incomplete data. As a result, our final study population consisted of 1,730 males and 1,732 females.

### Data source and data collection

The NHS South Sudan 2020 comprised nine questionnaires. However, in this study, we worked exclusively with the questionnaire "Domestic and Gender-based Violence," which was

adapted from WHO's multi-country study on women's health and domestic Violence Against Women 2005 [4,35]. This questionnaire was provided by United Nations Children`s Fund (UNICEF) subcluster in September 2020 [23,35].

The questionnaire collected information about the attitudes towards IPV and other forms of VAWG, as well as sociodemographic characteristics (gender, age, marital status, residence and educational level) among men and women aged 15-49 years old [23,35].

Interview data were collected in the local language between November 2020 and February 2021 by a team of 104 data collectors and 13 supervisors, trained by the Liverpool School of Tropical Medicine (LSTM) the week before the data collection began [35]. Translated instruments were backtranslated into English to ensure that the original meaning of the question was retained. The government provided electronic and hard copies of county maps used to locate the villages and bomas selected PPS to survey.

Data were collected using smart phones programmed with Open Data Kit (ODK) [43]. Supervisors were responsible for checking inconsistencies (duplications, erroneous entries) or missing data daily. If any of these issues were identified once data were uploaded to our server, the supervisor immediately assisted the interviewer to fix the error and/or obtain the missing data [23,35].

The mobile devices used for data collection were encrypted and password protected. Only members of the team knew the passwords. After uploading data, LSTM stored the data and created a password-protected database to which only the approved research team had access [23,35].

## Outcomes and exposures

We assessed the attitudes towards four forms of VAWG. In order to measure attitudes towards IPV we used the question: "Is a husband justified in hitting or beating his wife in the following five situations: i) goes out without telling, ii) neglects children, iii) argues with him, iv) refuses sex, and v) burns food" [35]. The possible responses were: acceptance, rejection and do not know. Any interviewee accepting any of these five scenarios scored 1 and 0 for a negative answer to all five sub-questions. Since the responses to these questions were binary, "do not know" was regarded as missing data, and the responder was given the state's median value [23]. In the next step, we measured attitudes towards the other 3 forms of VAWG, using the following question: "Would it be justified … to accept or practice any of the following: i) marrying off a child below the age of 18 years, ii) raiding villages for women during cattle rustling or other raids, and iii) female genital mutilation" [35]. Interviewees accepting at least one of these three options scored 1; otherwise, they received a 0 [23]. In the last step, we created a variable, "overlapping attitudes", identifying individuals who accepted both IPV and at least one additional form of VAWG. Interviewees scoring 1 on both attitudes scored 1 whilst interviewees who rejected one or both sets of practices, scored 0.

We selected exposure variables based on previous research [8,27,44] and their availability in the NHS South Sudan 2020. We classified variables at the individual, county and regional levels. The individual level included age, marital status (never married; married, cohabited or living together; and widowed, divorced or separated), and educational level (illiteracy; primary; and secondary or upper). The county level included collective community educational achievement, which was created using an index defined as follow: if over 50% of the interviewers in a county had any level of education, the county was classified as "literate", otherwise, "illiterate". Two counties (Malakal and Torit) having the same percentage (50%) of people literate and illiterate, we classified as "illiterate" based on the number of people and level of severity of education needs reported by United Nations in 2020 [20].

At the regional level, we grouped states into three categories: greater Equatoria (Western Equatoria, Central Equatoria and Eastern Equatoria), greater Upper Nile (Jonglei, Unity and Upper Nile), and greater Bahr el Ghazal (Northern Bahr el Ghazal, Western Bahr el Ghazal, Lakes, and Warrap) [35].

## Data analysis

**Prevalence of attitudes towards IPV and other forms of VAWG.** Data preparation was conducted using SPSS Statistics 28.0.1.0. For the data analysis, we used R version 4.1.2 (2021-11-01) to estimate the prevalence (with 95% confidence intervals) of attitudes towards IPV, child marriage, raiding villages for women during cattle rustling or other reasons for raids, female genital mutilation, and the overlapping acceptance of IPV and at least one other form of VAWG.

**Spatial analysis of overlapping attitudes towards IPV and at least one other form of VAWG.** To assess the distribution of the overlapping acceptance of IPV and at least one other form of VAWG in the country and its randomness [30], we estimated Global Moran's I using ArcGIS v10.8. This spatial statistic estimates spatial autocorrelation of a variable, with values ranging between -1 and +1, where 0 indicates no spatial pattern, a positive value indicates spatial autocorrelation (clustering), and a negative value indicates dispersion [25,45].

We also conducted a hot spot analysis to identify local clustering among the counties using the Getis and Ord's local Gi* statistic as a local indicator of spatial autocorrelation, implemented in ArcGIS v10.8. The Gi* statistic identifies clusters with strong positive or negative correlations ("hotspots" and "coldspots", respectively) in spatial data by contrasting local estimates of spatial autocorrelation with global averages [46]. The false discovery rate was applied to account for multiple testing corrections [47].

We then used Kulldorff's SaTScan software to identify statistically significant clusters of overlapping acceptance of IPV and at least one other form of VAWG by applying the Bernoulli purely spatial model, in which people who accepted both IPV and at least one additional form of VAWG were cases and people who did not were controls. Clusters were detected by p-values and likelihood ratio tests (LLR) using 999 Monte Carlo replications [48]. The highest LLR suggests a higher likelihood of being a cluster [49].

**Multilevel regression of overlapping attitudes towards IPV and at least one other form of VAWG.** Initially, we conducted a multicollinearity test to assess collinearity among the predictors using the variance inflation factor (VIF) [50], which indicated no collinearity among them. Thereafter, we fitted a multilevel logistic regression model to identify variables associated with the overlapping acceptance of IPV and at least one other form of VAWG at the individual, county, and regional levels, while accounting for cluster effects. In our models, male and female interviewees were nested within counties, which served as the clusters [27].

We considered clusters as a random effect to understand the unexplained variability at the county level. To do this, we fitted four models. Firstly, Model 0 did not contain predictors (random intercept). In Model I we included only individual level factors. Model II only included factors at the county and regional level, and Model III included individual, county and regional factors [25,27,50]. Odds ratios and 95% CI were estimated for all models using R package lme4 [51].

We used the likelihood ratio test (LLR) to compare model fit, with the best-fit model having the highest LLR [50]. Finally, further sensitivity analyses are presented in the S1 Text.

## Ethical considerations

The NHS 2020 South Sudan study received approval from the Ministry of Health in South Sudan and the internal ethics committee of the LSTM [35]. The latter determined that this

secondary analysis did not require an additional ethical review, as it was nested within a larger study that had already obtained approval from both the institutional research committee at LSTM and the local Ministry of Health.

In the original survey, participants provided informed consent. This consent was documented through signed agreements. For participants who were illiterate, data collectors offered to read the consent form to them, and instead of signing, participants used their thumbprints to indicate their agreement [35]. Data were anonymised once data reliability had been established.

## Results

### Characteristics of the participants

The final sample size was 3,462 (1,730 males and 1,732 females); 18 interviewees from county Nyirol were excluded due to incomplete surveys resulting from insecurity during the data collection. The average interviewee's age was 30.28 (SD = 9.40). Over two-thirds (78.02%) of the population were married, cohabited or living together; 56.27% of the population were illiterate. At the county level, 55.75% of the population lived in predominantly illiterate counties; 41.77% of the population lived in the Greater Upper Nile region (Table 1).

### Prevalence of attitudes towards IPV and other forms of VAWG

Of the attitudes towards the four types of VAWG studied, acceptance of IPV had the highest prevalence (71.50%, 95% CI = 69.99%–73.00%). Of the five IPV scenarios assessed, "being beaten for neglecting children" was the most accepted, while "being beaten for burning food" was the least accepted (Table 2).

Among the other three types of VAWG, acceptance of child marriage was the highest (37.98%, 95% CI = 36.37%–39.60%), although it was lower than the acceptance of IPV, which had the

**Table 1.  Sociodemographic characteristics of the participants.**

| Variable | Frequency (n=3,462) | Percentage |
|---|---|---|
| **Age** | 30.28 (Mean) | 9.40 (SD) |
| **Gender** | | |
| Men | 1,730 | 49.97 |
| Women | 1,732 | 50.03 |
| **Marital Status** | | |
| Never married | 601 | 17.36 |
| Married, cohabited or living together | 2,701 | 78.02 |
| Widowed, divorced or separated | 160 | 4.62 |
| **Educational level** | | |
| Illiterate | 1,948 | 56.27 |
| Primary | 940 | 27.15 |
| Secondary or upper | 574 | 16.58 |
| **Community literacy level** | | |
| Illiterate | 1,930 | 55.75 |
| Literate | 1,532 | 44.25 |
| **Region** | | |
| Greater Equatoria | 912 | 26.34 |
| Greater Upper Nile | 1,446 | 41.77 |
| Greater Bahr el-Ghazal | 1,104 | 31.89 |

**Table 2. Prevalence of attitudes accepting VAWG.**

| Attitudes accepting VAWG | Prevalence (%) | 95% CI |
|---|---|---|
| **Attitude accepting IPV** | 71.50 | 69.99-73.00 |
| To be beaten for going out without husband's permission. | 54.42 | 52.74-56.09 |
| To be beaten for neglecting children. | 55.71 | 54.04-57.38 |
| To be beaten for arguing with husband. | 45.61 | 43.94-47.28 |
| To be beaten for refusing have sex. | 35.64 | 34.05-37.27 |
| To be beaten for burning food. | 30.67 | 29.15-32.25 |
| **Attitude accepting child marriage** | 37.98 | 36.37-39.60 |
| **Attitude accepting raiding villages for women during cattle rustling or other raids** | 14.67 | 13.50-15.90 |
| **Attitude accepting female genital mutilation** | 4.22 | 3.55-4.90 |
| **Overlap between attitudes accepting IPV and at least one other form of VAWG** | 34.72 | 33.13-36.30 |

highest prevalence (71.50%, 95% CI = 69.99%–73.00%). Female genital mutilation was the least accepted (4.22%, 95% CI = 3.55%–4.90%) (Table 2). The prevalence of the overlap between IPV acceptance and at least one other form of VAWG was 34.72% (95% CI = 33.13%–36.30%).

## Global and local spatial analysis of overlapping attitudes towards IPV and at least one other form of VAWG

The Global Moran's I for the overlap between acceptance of IPV and least one other form of VAWG was positive (0.23) and statistically significant (p-value < 0.01), indicating a non-random correlation and a clustering effect (S1 Fig). The Getis and Ord's Local Gi* identified hotspots of this overlapping at the 90% level of confidence in the counties Kapoeta East, Kapoeta North, and Kapoeta South as well as in Mayendit and Leer (Fig 1).

## Sat Scan analysis of overlapping attitudes towards IPV and at least one other form of VAWG

A Sat Scan analysis identified six significant clusters of overlapping acceptance of IPV and at least one other form of VAWG (Fig 2). The primary clusters were located in the counties of Kapoeta East, Kapoeta South, Kapoeta North, Budi, and Ikotos, all within Eastern Equatoria state, and Pibor (located in the Greater Pibor Administrative Area, which was merged with Jonglei state in our spatial analysis). The most notable cluster was centred at 5.138396°N, 34.652377°E, with a radius of 208.80 km (relative risk = 2.71; log-likelihood ratio = 151.69; p < 0.001). Consequently, individuals living within this cluster have a 2.71 times higher risk of exhibiting an overlapping acceptance of IPV and at least one other form of VAWG compared to those living outside it (Table 3).

The main secondary cluster (log-likelihood ratio = 119.63) included the counties of Mayendit, Leer, Koch, and Guit, all located in Unity State. This cluster was centred at 8.633362°N, 29.856306°E, with a radius of 67.00 km (relative risk = 2.89). Consequently, individuals living within this cluster have a 2.89 times higher risk of exhibiting overlapping acceptance of IPV and at least one other form of VAWG compared to those residing outside this area.

## Multilevel logistic regression of overlapping attitudes towards IPV and at least one form of VAWG

At the individual level (Model I), only marital status displayed an association with the overlap. People married, cohabiting or living together (aOR = 1.40, 95% CI = 1.04-1.90), and people

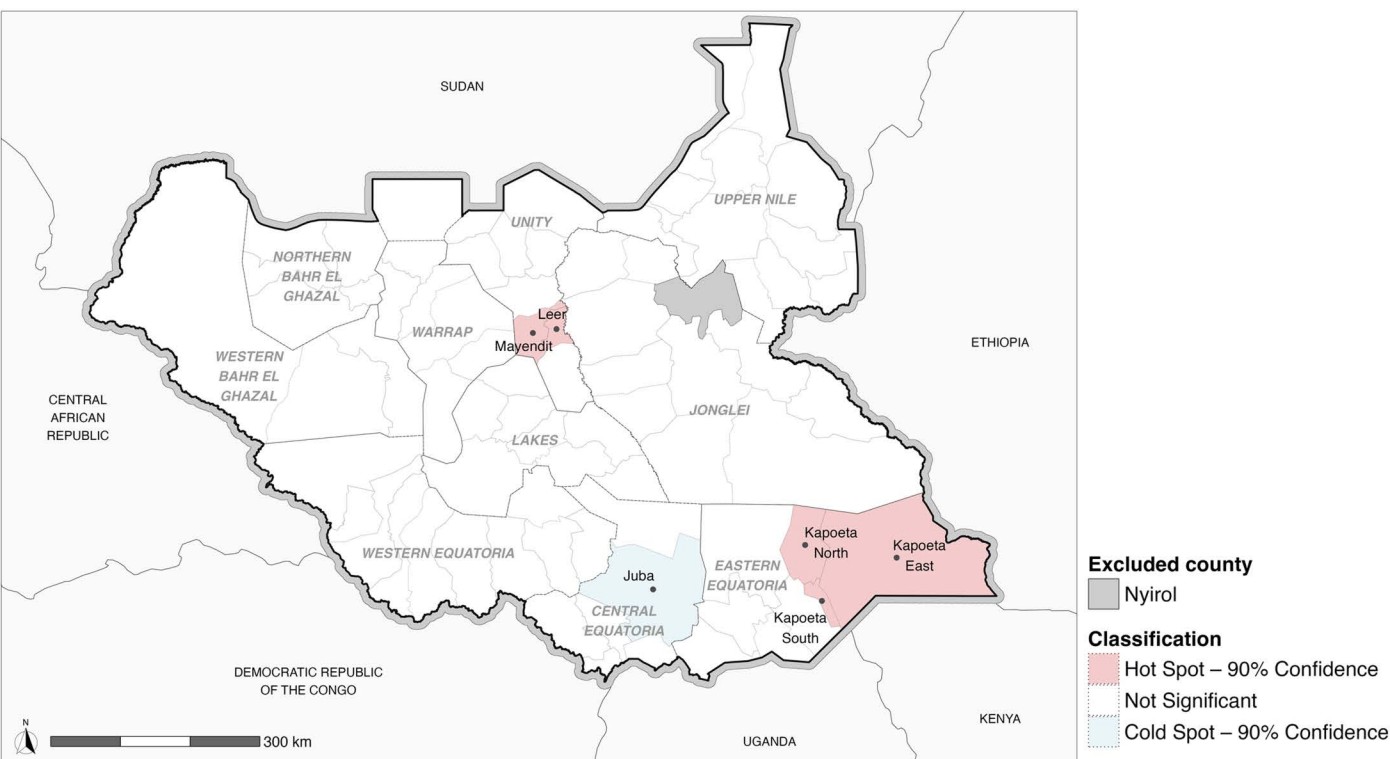

**Fig 1. Hotspot analysis of the overlap of IPV acceptance and at least one other form of VAWG. Source: South Sudan Information Management Working Group (IMWG), National Bureau of Statistics (NBS), International Organization for Migration (IOM) and United Nations Office for the Coordination of Humanitarian Affairs (OCHA)** https://data.humdata.org/dataset/cod-ab-ssd).

widowed, divorced or separated (aOR = 1.75, 95%CI = 1.05-2.93) had a higher odds of revealing overlapping acceptance IPV and at least one other form of VAWG compared to people who never married (Table 4).

At a county and regional level (Model II), people who live in counties with the majority population being literate had a lower odds of overlapping acceptance IPV and at least one other form of VAWG (aOR = 0.24, 95% CI = 0.09-0.63) compared to predominately illiterate counties.

Model III included individual, county, and regional level variables for marital status (married, cohabiting or living together, widowed, divorced, or separated), which showed higher odds of overlapping acceptance of IPV and at least one other form of VAWG compared to those who were never married. Specifically, the adjusted odds ratios were 1.40 (95% CI = 1.03–1.90) for cohabiting or living together, and 1.76 (95% CI = 1.05–2.93) for widowed, divorced, or separated individuals. Conversely, people residing in counties where the majority of residents have some level of education had lower odds of overlapping acceptance of IPV and other forms of VAWG (aOR = 0.26, 95% CI = 0.09–0.69) compared to those in counties where the majority of people are illiterate.

Regarding the random effect, for Model 0, the county variance was 5.12 (95% CI = 3.47–7.60), indicating considerable variation among different county clusters in the likelihood of interviewees displaying overlapping acceptance of IPV and at least one other form of VAWG. This suggests that factors specific to each cluster may account for this overlap [52]. Additionally, 61% of the variation in this overlap was attributed to variation between clusters

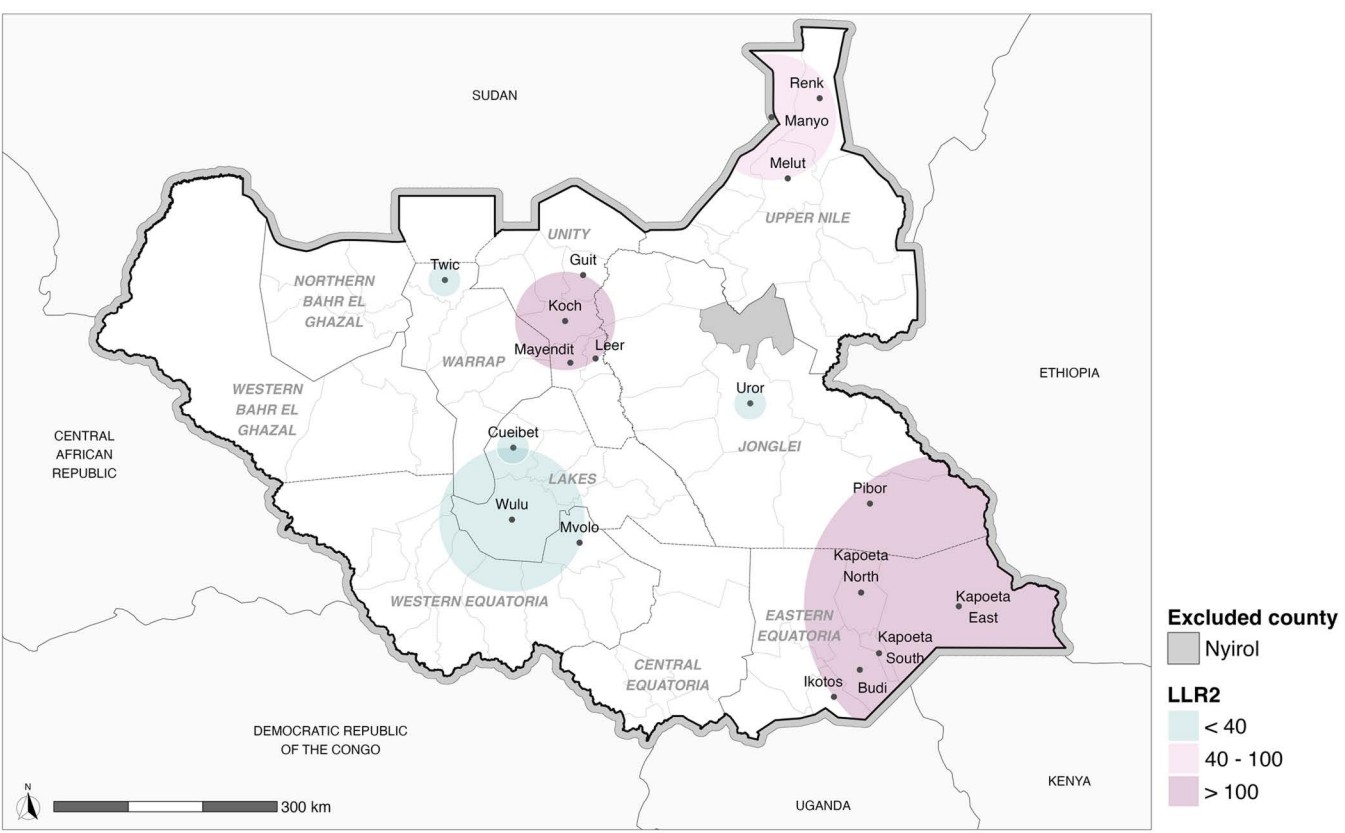

**Fig 2. SatScan analysis map of overlapping acceptance of IPV and at least one other form of VAWG. Source: South Sudan Information Management Working Group (IMWG), National Bureau of Statistics (NBS), International Organization for Migration (IOM) and United Nations Office for the Coordination of Humanitarian Affairs (OCHA)** https://data.humdata.org/dataset/cod-ab-ssd).

Table 3. SatScan analysis results of clusters of overlapping between acceptance IPV and at least one other form of VAWG.

| Location(s) | Coordinates/ radius | Popu-lation | Cases | Relative Risk | Log likeli-hood ratio | p-value |
|---|---|---|---|---|---|---|
| **Kapoeta East, Kapoeta South, Kapoeta North, Budi, Pibor, Ikotos** | 5.138396 N, 34.652377 E/ 208.80 km | 286 | 236 | 2.71 | 151.69 | <0.001 |
| **Mayendit, Leer, Koch, Guit** | 8.633362 N, 29.856306 E/ 67.00 km | 152 | 141 | 2.89 | 119.63 | <0.001 |
| **Manyo, Renk, Melut** | 11.131136 N, 32.370237 E/ 86.13 km | 114 | 88 | 2.32 | 44.52 | <0.001 |
| **Wulu, Mvolo, Cuibet,** | 6.200243 N, 29.208889 E/ 98.03 km | 114 | 64 | 1.65 | 11.26 | <0.001 |
| **Uror** | 7.625310 N, 32.111289 E/ 0 km | 38 | 35 | 2.70 | 28.20 | <0.001 |
| **Twic** | 9.135899 N, 28.391979 E/ 0 km | 38 | 29 | 2.23 | 13.88 | <0.001 |

(intraclass correlation coefficient = 0.61), while only 39% was attributed to variation within clusters [46,50].

A similar trend was observed in Model I. The variation among clusters slightly decreased to 56% in Models II and III. However, as Model III displayed the highest log-likelihood, it was considered the best fit for the data. Consequently, we used the variables included in Model III to discuss the predictors of overlapping acceptance of IPV and at least one other form of VAWG in South Sudan.

**Table 4. Multilevel logistic regression (aOR, 95% CI) of overlapping acceptance IPV and at least one other form of VAWG.**

| Characteristics | Model 0 | Model I aOR[†] (95% CI) [p-value] | Model II aOR[†] (95% CI) [p-value] | Model III aOR[†] (95% CI) [p-value] |
|---|---|---|---|---|
| **Age** | | 0.99 (0.98-1.00) [0.18] | | 0.99 (0.98-1.00) [0.18] |
| **Gender** | | | | |
| Men | | Ref | | Ref |
| Women | | 1.03 (0.84-1.26) [0.80] | | 1.03 (0.84-1.27) [0.76] |
| **Marital Status** | | | | |
| Never married | | Ref | | Ref |
| Married, cohabited or living together | | 1.40 (1.04-1.90) [0.03] | | 1.40 (1.03-1.90) [0.03] |
| Widowed, divorced or separated | | 1.75 (1.05-2.93) [0.03] | | 1.76 (1.05-2.93) [0.03] |
| **Education level** | | | | |
| Illiteracy | | Ref | | Ref |
| Primary | | 0.87 (0.67-1.13) [0.30] | | 0.90 (0.70-1.16) [0.40] |
| Secondary or upper | | 0.82 (0.60-1.12) [0.20] | | 0.84 (0.62-1.14) [0.27] |
| **Community literacy level** | | | | |
| Illiteracy | | | Ref | Ref |
| Non-Illiteracy | | | 0.24 (0.09-0.63) [<0.01] | 0.26 (0.09-0.69) [<0.01] |
| **Region** | | | | |
| The Greater Equatoria | | | Ref | Ref |
| The Greater Upper Nile | | | 2.39 (0.75-7.63) [0.15] | 2.41 (0.75-7.72) [0.15] |
| The Greater Bahr El-Ghazal | | | 0.88 (0.25-3.09) [0.85] | 0.91 (0.26-3.20) [0.89] |
| **Random effects results** | | | | |
| County Variance (95% CI) | 5.12 (3.47-7.60) | 5.04 (3.41-7.45) | 4.23 (2.85-6.27) | 4.23 (2.85-6.26) |
| Interclass correlation coefficient (ICC) | 0.61 | 0.61 | 0.56 | 0.56 |
| **Model fitness** | | | | |
| Log-likelihood | -1551.3834 | -1545.7192 | -1545.1118 | -1539.8757 |

[†]aOR= adjusted Odds ratio.

## Discussion

### Prevalence of attitudes accepting VAWG

In our study, the national South Sudanese prevalence among women and men aged 15-49 years with attitudes accepting IPV was 71.50% (95% CI = 69.99%-73.00%), which is substantially higher than the average prevalence in low- and middle-income countries [53], and in the sub-Saharan and North Africa regions (41.09%, 95% CI = 31.71%-36.40% and 44.95%, 95% CI = 31.85%-38.40%, respectively). Although there was a decrease of nearly 8% in women's acceptance of IPV between the South Sudan national household surveys of 2010 [54] and 2020, the prevalence of accepting IPV in the country is still one of the highest worldwide [15].

Regarding the other three forms of VAWG assessed (child marriage, raiding villages for women during cattle rustling or other raids, and female genital mutilation), child marriage had the highest prevalence of acceptance (37.98%, 95% CI = 36.37%-39.60%). It is worth noting that 51.5% of women aged 20-24 years were married before age 18. In some states, such as Unity, 71% of girls were married before age 18, and 10% before age 15 [55]. According to the UNICEF, South Sudan has the seventh-highest prevalence of child marriage worldwide [16].

The prevalence of overlapping acceptance of IPV and at least one other form of VAWG was 34.72% (95% CI = 33.13%-36.30%). The most frequent overlapping attitude with IPV was child marriage. In our previous research [23], both forms of VAWG shared drivers such as a low level of education or residing in places with a high level of violent conflicts or with internally displaced people. Despite that, the high prevalence of attitudes accepting VAWG in South Sudan is far from being explained exclusively by these drivers. A more in-depth analysis requires an understanding of multiple factors rooted in a history of gender inequalities, economic crises, cultural factors and humanitarian needs in South Sudan [5] which are beyond the scope of this paper.

### Spatial analysis of overlapping attitudes accepting IPV and at least one other form of VAWG

In the spatial analysis, the main cluster of "overlapping attitudes" was located in Kapoeta East, Kapoeta North, Kapoeta South, Budi, Ikotos (Eastern Equatoria state), and Pibor (Greater Pibor Administrative Area). Attitudes accepting IPV were highly prevalent across the country, underscoring the importance of analysing additional local factors that increase the acceptance of it overlapping with other forms of VAWG.

For instance, Kapoeta East, Kapoeta North, Kapoeta South, and Budi used to be one county. Hence, the culture, social norms and practices are similar. Ethnically and culturally, the Toposa people are the primary inhabitants of Kapoeta East, Kapoeta North, and Kapoeta South [56], where nomadic-pastoralism is the predominant way of life [57]. In contrast, Budi is mainly inhabited by the Didinga and Buya tribes, who primarily practice crop agriculture [56]. Although this study has not directly linked the observed clustering of acceptance of IPV and at least one other form of VAWG to culture and cultural practices in these locations, the Toposa's nomadic pastoralist way of life is particularly associated with cattle raiding and intercommunal violence, which previous studies have linked to the normalisation of VAWG [58].

In Pibor, practices of raiding children and women during cattle raids, inter-communal violence, and inter-generational fighting are routine occurrences [57]. Ikotos has insecure conditions due to cattle raids, conflict between government and armed groups, and displacement [57].

Cattle raiding and intercommunal violence have been associated with VAWG in conflicted settings, such as South-Sudan [8,9]. Cattle raiding used to escalate to intercommunal violence [21,24] can lead to women and girls being targeted (e.g., non-partner sexual violence,

abduction, etc) [9,23]. Additionally, intercommunal violence could reinforce hyper masculine behaviours and patriarchal norms and practices such as normalise social acceptance of IPV and child marriage [9,23,53]. These dynamics could explain the clustering effect observed in our study.

## Predictors of overlapping attitudes accepting IPV and at least one other form of VAWG

Model III (Table 4) contains individual, county and regional level variables and exhibited the best goodness of fit with the data. Accordingly, 56% of the variance in the overlap in attitudes accepting IPV and other expressions of VAWG is attributed to variation between clusters and 44% is attributed to variations at the cluster level (i.e., differences among participants within the same county).

Model III identified two factors associated with overlapping acceptance attitudes: marital status and community literacy level. People who were married, cohabiting, or living together had higher odds (aOR = 1.40, 95% CI = 1.03-1.90) of displaying overlapping attitudes. Similarly, people who were widowed, divorced, or separated compared to those who had never married exhibited higher odds of overlapping acceptance of IPV and at least one other form of VAWG (aOR = 1.76, 95% CI = 1.05-2.93). In an analysis by gender, only women maintained this association.

These findings are consistent with previous studies in low- and middle-income countries, which indicate that women who have never been married are more likely to reject gender inequities, such as VAWG, compared to women of the same age and socioeconomic status who have ever partnered [59].

In order to explain this phenomenon, we should understand the link between marriage and income in South Sudanese society. Usually, the groom's family must pay a dowry to the bride's family, which is generally cattle, the country's main financial resource [5]. As a result, other researchers conclude that the existing intercommunal violence together with poverty have led to raiding villages for women and girls to get a wife, thereby avoiding paying a dowry [5,12,24]. Families could also force their daughters to marry to receive an income [16,55].

This dynamic identifies women and girls as objects of transaction, resulting in women recognising their partners as owners [5,16], which could decrease their perceived right to reject any act of violence against them [60]. As a result, ever-married people could develop attitudes accepting VAWG, aligning with predominant community attitudes and social norms. Conversely, those who never married could value gender equality and women's empowerment, which may encourage them to challenge traditional patriarchal norms that support VAWG.

Despite these observations in South Sudan, researchers do not agree on whether marital status increases the risk of or offers protection against VAWG, especially IPV [8]. Some theories claim that marriage may offer some protection compared to cohabiting. However, others suggest that being married increases the risk of sexual violence by the husband compared to those who cohabit or live together [8].

On the other hand, community literacy level also displayed an association with attitudes accepting VAWG. Counties where the majority of the population has some level of education showed lower odds of overlapping acceptance of IPV and at least one other form of VAWG (aOR = 0.26, 95% CI = 0.09-0.69) compared to counties where the majority of the population is illiterate. In our study, education played a mitigating role to VAWG by shaping individual attitudes and those of others residing in the county [61,62].

Education at the community and social levels allows the participation of women in activities outside of their gender role and increases their decision-making capacities [7], which

promotes a change in attitudes away from accepting VAWG [55], such as blaming the victims, lack of social and legal support, and maltreatment in health care services [63].

However, the World Bank, in 2018, reported that in South Sudan, only 35% of the population aged 15 and above were literate with men showing a higher literacy rate (40%) than women (29%) [64]. Factors such as climate change, war, and displacement [21] have negatively impacted children's schooling, especially among boys [65]. However, this influence on literacy could be underestimated among girls, because they already experienced previous social and cultural barriers to access schooling [5,55,65].

## Strengths and limitations

To our knowledge, this is the first study applying spatial analysis and multilevel regression analysis to VAWG in South Sudan at a local level. Therefore, our findings demonstrated how our approach complements previous studies in the country [5,23]. Due to our large national sample using PPS, these data, which have good geographic coverage may be generalisable to other settings.

However, this study has some limitations. For example, our instruments did not allow us to capture the data based on traditional categories in VAWG, such as physical, sexual or psychological violence. Furthermore, in order to sample in each county using PPS, we used the sampling frames resulting from a projection of the 2008 census by the South Sudan National Bureau of Statistics which is the last available census in the country. Therefore, these projections could be inaccurate due to population movement associated with internal displacement of people, returnees, and refugees to neighbouring countries.

Also, as our data were collected in 2020, it may not necessarily represent the current country situation. Another limitation is that because our spatial analysis was at a county level, it would not allow the identification of potential cluster effects at the sub-county level (for example, villages). Additionally, each county may have unique factors that influence their similarities or differences which we did not measure and warrant further research. We did not analyse data disaggregated by gender, which neither would allow for identifying cluster effects by gender.

## Conclusions and recommendations

We analysed attitudes accepting four types of VAWG (IPV, child marriage, raiding villages for women during cattle rustling or other raids, and female genital mutilation) and the overlapping acceptance of IPV and at least one other form of VAWG. The most frequent overlapping attitudes were accepting both IPV and child marriage.

These overlapping attitudes are non-randomly distributed. The areas identified as primary clusters (counties located in Kapoeta East, Kapoeta North, Kapoeta South, Budi, Pibor and Ikotos) require focused interventions to decrease the prevalence of attitudes supporting VAWG. As the results of our different spatial techniques (Ord G and SatScan) are quite similar, the identified clusters of overlapping acceptance IPV and at least one other form of VAWG could be treated as valid.

People married, cohabited or living together as well as people widowed, divorced or separated were associated with overlapping attitudes accepting IPV and at least one other form of VAWG. Conversely, community literacy level was associated with attitudes rejecting them.

Attitudes accepting VAWG should not be considered the only factor stimulating VAWG. Rather, they should be analysed within a broader framework. We recommend further investigation in the hotspot clusters to better understand the predictors of attitudes accepting VAWG. Also, as 44% of the variance in the overlapping acceptance of IPV and other forms of VAWG is attributed to variations within clusters, we suggest conducting qualitative research

to investigate this phenomenon. That research should also use the socioecological framework to understand drivers of VAWG [9] and investigate the impact the history of conflict in South Sudan has had on VAWG. This information can further inform policy formation, programme planning, and implementation aimed at ameliorating these attitudes.

To achieve this goal, it is essential to increase access to education for children and address social barriers supported by gender inequities or biased ideas about gender roles that prevent or discourage education. Additionally, promoting gender equity among adults through women's empowerment and human rights initiatives can strengthen the healthcare system's ability to manage cases of VAWG effectively.

Additionally, the spatial analysis of the overlapping acceptance of IPV and at least one other form of VAWG could aid policy makers and the Ministry of Health to prioritise funding to implement strategies to tackle VAWG. This contribution is fundamental to South Sudan's management of prevention, mitigation and case management interventions which almost exclusively depends on non-governmental organisations supported by international donors [66,67].

## Supporting information

**S1 Table. Sample size estimation.** Description: For each state of the number of supervision areas (SAs), number of interviews per SA, total sample size, and maximum 95% Confidence Interval.
(DOCX)

**S1 Text. Sensitivity analysis.** Description: Sensitivity analysis of the prevalence, spatial analysis results and predictors of attitudes accepting both intimate partner violence and additional forms of violence against women and girls in South Sudan.
(DOCX)

**S1 Fig. Description: Spatial autocorrelation of the overlap of IPV acceptance and at least one other form of VAWG. Results of the Global Moran´s I analysis.**
(TIF)

## Acknowledgments

We are grateful to Dr Susan Bartels, Canada Research Chair in Humanitarian Health Equity at Queen's University, Canada for her priceless feedback and support. The authors want to acknowledge the Ministry of Health of the Republic of South Sudan and UNICEF South Sudan without whose support this research would not have been possible.

## Author contributions

**Conceptualization:** Angelo Lamadrid, Ignacio Leiva-Escobar, Richard Lako, Joseph J. Valadez.

**Data curation:** Caroline Jeffery, Robert J. Anguyo, Joseph J. Valadez.

**Formal analysis:** Angelo Lamadrid, Ignacio Leiva-Escobar, Caroline Jeffery.

**Funding acquisition:** Richard Lako, Joseph J. Valadez.

**Investigation:** Angelo Lamadrid, Ignacio Leiva-Escobar, Caroline Jeffery, Joseph J. Valadez.

**Methodology:** Angelo Lamadrid, Ignacio Leiva-Escobar, Caroline Jeffery, Robert J. Anguyo, Joseph J. Valadez.

**Project administration:** Joseph J. Valadez.

**Resources:** Joseph J. Valadez.

**Software:** Angelo Lamadrid, Ignacio Leiva-Escobar.

**Supervision:** Joseph J. Valadez.

**Validation:** Joseph J. Valadez.

**Visualization:** Angelo Lamadrid, Ignacio Leiva-Escobar.

**Writing – original draft:** Angelo Lamadrid, Ignacio Leiva-Escobar, Caroline Jeffery, Robert J. Anguyo, Richard Lako, Joseph J. Valadez.

**Writing – review & editing:** Angelo Lamadrid, Ignacio Leiva-Escobar, Caroline Jeffery, Robert J. Anguyo, Richard Lako, Joseph J. Valadez.

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
