## [Decision Letter · Decision Letter 0]

4 Dec 2024

PGPH-D-24-01742

Prevalence, locations and predictors of attitudes accepting both intimate partner violence and additional forms of violence against women and girls in South Sudan: a geospatial analysis

Dear Dr. Valadez,

Thank you for submitting your manuscript to PLOS Global Public Health. After careful consideration, we feel that it has merit but does not fully meet PLOS Global Public Health’s publication criteria as it currently stands. Therefore, we invite you to submit a revised version of the manuscript that addresses the points raised during the review process.

Editor comments:

The reviewers and I found the manuscript to be well-written and provide important insights into spatial factors associated with gender-based violence in South Sudan. Given the ongoing humanitarian crises in South Sudan, and the difficulties in conducting research in this setting, the manuscript fills an important gap.However, please note areas where the manuscript could be improved. Specifically, the reviewers raised several questions concerning the sampling approach and methodology that need to be clarified. Consider whether all tables and figures are appropriately formatted and integrated with the text. Finally, consider the suggestions regarding the discussion, particularly in terms of whether the literature referenced could be updated, and whether you might be able to better situate the findings of your study in the context of the literature on gender-based and humanitarian crises.

We look forward to receiving your revised manuscript.

Kind regards,

Marie A. Brault, PhD

Academic Editor

Journal Requirements:

Additional Editor Comments (if provided):

Reviewers' comments:

Reviewer's Responses to Questions

**Comments to the Author**

1. Does this manuscript meet PLOS Global Public Health’s publication criteria ? Is the manuscript technically sound, and do the data support the conclusions? The manuscript must describe methodologically and ethically rigorous research with conclusions that are appropriately drawn based on the data presented.

Reviewer #1: Yes

Reviewer #2: Yes

Reviewer #3: Yes

2. Has the statistical analysis been performed appropriately and rigorously?

Reviewer #1: Yes

Reviewer #2: Yes

Reviewer #3: Yes

3. Have the authors made all data underlying the findings in their manuscript fully available (please refer to the Data Availability Statement at the start of the manuscript PDF file)?

Reviewer #1: Yes

Reviewer #2: No

Reviewer #3: Yes

4. Is the manuscript presented in an intelligible fashion and written in standard English?

Reviewer #1: Yes

Reviewer #2: Yes

Reviewer #3: Yes

5. Review Comments to the Author

Reviewer #1: Its a suggestion to amend the conclusion by describing only the variable (married women) as the most high likely ratio of associated with IPV. 11 % of the references were old than 10 years. try to replace them with new ones.

Reviewer #2: The research needs major modifications and some important components of the research has been missed. So, try to revise the manuscript based on the provided comments. In addtion, follow the Plos Global Public Health journal policies on how to prepare manuscripts.

Reviewer #3: Dear Authors,

Thank you for your contribution to PLOS Global Health. I want to commend you for choosing a salient topic and appropriately situating, analyzing, and presenting the spatial analyses in a compelling manner.

Please see the attached PDF for my comments.

Best,

Reviewer

6. PLOS authors have the option to publish the peer review history of their article (what does this mean? ). If published, this will include your full peer review and any attached files.

**Do you want your identity to be public for this peer review?** For information about this choice, including consent withdrawal, please see our Privacy Policy .

Reviewer #1: **Yes: ** Hadiqa Adnan

Reviewer #2: **Yes: ** First Name: Habtamu Molla

Last Name: Ayele

Affiliation: Maternal and Child Health Directorate, Federal Ministry of Health, Addis Ababa, Ethiopia.

Reviewer #3: **Yes: ** Jacob Michael Souch, B.A., B.S.

---

## [Editor Report · Decision Letter 1]

20 Feb 2025

Prevalence, locations and predictors of attitudes accepting both intimate partner violence and additional forms of violence against women and girls in South Sudan: a geospatial analysis

PGPH-D-24-01742R1

Dear Professor Valadez,

We are pleased to inform you that your manuscript 'Prevalence, locations and predictors of attitudes accepting both intimate partner violence and additional forms of violence against women and girls in South Sudan: a geospatial analysis' has been provisionally accepted for publication in PLOS Global Public Health.

Best regards,

Marie A. Brault, PhD

Academic Editor